# Imbalanced Fault Diagnosis of Rotating Machinery Based on Deep Generative Adversarial Networks with Gradient Penalty

**Junqi Luo** [1,2] **, Liucun Zhu** [1,*]**, Quanfang Li** [3]**, Daopeng Liu** [3] **and Mingyou Chen** [1]

1   Advanced Science and Technology Research Institute, Beibu Gulf University, Qinzhou 535000, China;
    2011401005@st.gxu.edu.cn (J.L.); chenmingyou@bbgu.edu.cn (M.C.)
2   College of Mechanical Engineering, Guangxi University, Nanning 530004, China
3   School of Mechanical Engineering, Jiangsu University, Zhenjiang 212000, China;
    2221903122@ujs.edu.cn (Q.L.); 2222003017@stmail.ujs.edu.cn (D.L.)
*   Correspondence: lczhu@bbgu.edu.cn

**Abstract:** In practical industrial application, the fault samples collected from rotating machinery are frequently unbalanced, which will create difficulties when it comes to diagnosis. Besides, the variation of working conditions and noise factors will further reduce the diagnosis's accuracy and stability. Considering the above problems, we established a model based on deep Wasserstein generative adversarial network with gradient penalty (DWGANGP). In this model, the unbalanced fault data set will first be trained by the sample generation network to generate synthetic samples, which will be used to restore the balance. A one-dimensional convolutional neural network with a specific structure is then used as the fault diagnosis network to classify the reconstructed equilibrium samples. The experimental results show that the proposed sample generation network can generate high-quality synthetic samples under highly imbalanced data, and the diagnostic network has a fast training convergence. Compared to the combination methods of support vector machines, back propagation neural network and deep belief network, our method has a 74% average accuracy in all unbalanced experimental conditions, which has 64%, 69% and 87% averages leading, respectively.

**Keywords:** convolutional neural network; fault diagnosis; generative adversarial networks; imbalance data

## 1. Introduction

Rolling bearings are widely used in industrial machinery, of which the health state has a significant influence on the performance and service life of the mechanical equipment. Because of the complex working environment, rolling bearing is one of the most vulnerable components in machinery. The faults of rolling bearings are not easy to be recognized visually; hence, the vibration signals collected by sensors are generally used for fault analysis [1,2].

In recent years, machine learning has emerged as a powerful tool in studying the fault diagnosis of machinery. Extensive research has shown that the K-means clustering, support vector machine (SVM), Bayesian network and multilayer perceptron (MLP) can be used in the field of mechanical fault diagnosis [3–6]. Such approaches, however, do not have sufficient ability to extract depth features of vibration signals, due to the limitation of their shallow network architectures. Therefore, relevant scholars try to use the signal processing methods such as Fourier transform (FFT), wavelet transform (WT) and empirical mode decomposition (EMD) to initially extract the feature of vibration signals in time- or frequency-domain [7–9]. Then, the refined signals are trained and classified by the machine learning methods. It is undeniable that the above scheme's diagnostic effect is often better than that of the single machine learning algorithm. However, it increases the complexity of the model, and the feature selection relies on manual labelling and expert knowledge, which is processing time-consuming.

Aiming at the above problems, various studies using deep neural networks have assessed the efficacy for fault diagnosis. Among them, autoencoder (AE), deep belief network (DBN) and convolutional neural network (CNN) have been shown to have the adaptive feature extraction capability of the raw vibration signals, which can provide end-to-end diagnostic solutions for mechanical faults [10–15]. Deep-learning-based methods provide a means of learning features and identifying faults automatically. Compared to the conventional machine learning algorithms, deep-learning methods can achieve high-accuracy fault diagnosis without a feature extractor.

Nonetheless, the greater part of the literature was based upon the assumption that all fault samples have the same probability distribution. So far, however, there has been little discussion about the fault samples are far less than the normal samples in machinery fault diagnosis. Recent studies have documented the oversampling techniques can improve the classified accuracy in imbalanced sample problem. Moreo et al. [16] used the random oversampling (NR) to accomplish the imbalanced text classification. Sun et al. [17] developed the synthetic minority oversampling technique (SMOTE), an interpolation method based on K nearest neighbor, to balance class-imbalanced financial data. Tan et al. [18] proposed a deep imbalance learning model with the adaptive synthetic sampling (ADASYN) and AE to assess the power system's transient stability under unstable samples. More recently, some studies also exploited the generative adversarial networks (GAN), and its variations, to settle the imbalance fault problem of machinery [19–21]. Wang et al. [22] combined the generative adversarial networks and stacked denoising autoencoders to the planetary gearbox fault diagnosis. Zhou et al. [23] design a GAN model to generate fault features rather than fault samples and be trained by the deep neural network to solve the unbalanced data problem. Shao et al. [24] considered an auxiliary classifier GAN-based framework for augmenting the unbalanced fault data of machine. Relevant literature has exemplified the effectiveness of the oversampling methods and GAN, but still faces the following limitations: (1) The synthetic samples produced by oversampling technique in highly unbalanced data were excessively similar, which leads to overfitting in training. (2) The GAN model may occur the vanishing gradient when the discriminator overpowers the generator, which trends to generate low-quality synthetic samples. (3) The GAN model may suffer from the mode collapse problem, resulting in an extremely low variety of the generated signals. (4) In the machinery fault diagnosis scenario based on imbalanced data, the variable working conditions and external noise factors are not taken into account in model performance evaluation.

To address the above deficiencies, we propose a rotating machinery fault diagnosis model for the imbalanced data problem named DWGANGP. The model comprises two independent networks: the sample generation network and the fault recognition network, respectively devised by Wasserstein generative adversarial network with gradient penalty (WGAN-GP) and CNN. In the diagnostic task, WGAN-GP first generates the high-quality synthetic samples from the minority samples (i.e., the unbalanced samples). These synthetic samples and the minority samples were extended to the majority samples (i.e., the balanced samples), and are trained and classified by the CNN. The main contributions are summarized as follows:

(1) Propose an end–end fault diagnosis model to optimize the imbalanced data problem of rotating machinery.

(2) Design the comparative experiments to verify the strong robustness of the proposed model under variant–load and noise conditions.

(3) Use Wasserstein distance as GAN's loss function and gradient penalty as GAN's training strategy, which be able to enhance the stability of GAN training and the quality of synthetic samples. As a result, our method provides a far better diagnostic accuracy than other models under highly unbalanced conditions.

The rest of this paper has been divided into five parts. Section 2 introduces the theoretical background of the generative adversarial network. Section 3 presents the

framework of the proposed model. Section 4 deals with the results of the case studies. Section 5 provides the discussion. Section 6 gives the conclusion.

## 2. Theoretical Network

### 2.1. Generative Adversarial Network

GAN [25] has an impressive performance in many generative tasks such as images and semantic segmentation. It is game theory-based and consists of two models: a generator G and a discriminator D, are competing and reinforcing each other, shown as Figure 1. A discriminator D distinguishes a sample from a given dataset blended real and false samples, aiming to try hard not to be cheated. The loss functions of the discriminator and the generator can be formulated as follows [25]:

$$L_D = \mathbb{E}_{x \sim p_r(x)}[\log D(x)] + \mathbb{E}_{z \sim p_z(z)}[\log(1 - D(G(z)))] \tag{1}$$

$$L_G = \mathbb{E}_{z \sim p_z(z)}[\log(1 - D(G(z)))] \tag{2}$$

where $p_r$ is the data distribution over real sample $x$. $p_z$ is the data distribution over noise input $z$. $D(x)$ represents the probability that $x$ comes from the real data. $\mathbb{E}_{x \sim p_r(x)}$ denotes the expectation of $x$ from the real distribution $p_r$. $\mathbb{E}_{z \sim p_z(z)}$ denotes the expectation of $z$ sampled from $p_z$. The discriminator wants to maximize $\mathbb{E}_{x \sim p_r(x)}[\log D(x)]$ but expects the $D(G(z))$ trend to zero by maximizing $\mathbb{E}_{z \sim p_z(z)}[\log(1 - D(G(z)))]$. Meanwhile, the generator wants $D(G(z))$ trend to one by minimizing $\mathbb{E}_{z \sim p_z(z)}[\log(1 - D(G(z)))]$. This zero-sum game between two models allows one to promote their functionalities steadily and eventually reach Nash equilibrium. Therefore, the global loss function should be optimized to the following expression:

$$\min_G \max_D L(D, G) = \mathbb{E}_{x \sim p_r(x)}[\log D(x)]$$
$$+ \mathbb{E}_{z \sim p_z(z)}[\log(1 - D(G(z)))] \tag{3}$$

### 2.2. Wasserstein Generative Adversarial Network with Gradient Penalty

Although GAN demonstrates an outstanding effect in the field of image generation, it still faces the dilemmas of instability and inefficiency in training. To alleviate the above problems, Arjovsky et al. [26] proposed WGAN using Wasserstein distance instead of the Jensen–Shannon (JS) divergence. Their expressions as shown follow separately [26].

$$W(p_r, p_g) = \inf_{\gamma \sim \Pi(p_r, p_g)} \mathbb{E}_{(x,y) \sim \gamma}[\|x - y\|] \tag{4}$$

$$D_{JS}(p \| q) = \frac{1}{2} D_{KL}(p \| \frac{p+q}{2}) + \frac{1}{2} D_{KL}(q \| \frac{p+q}{2}) \tag{5}$$

where $\Pi(p_r, p_g)$ represents the set of all simultaneous distribution $\gamma(x, y)$ between $p_r$ and $p_g$. $x$ and $y$ denote the real sample and synthetic sample from the above distribution. $\mathbb{E}_{(x,y) \sim \gamma}[\|x - y\|]$ denotes the expectation of distance between $x$ and $y$. $p$ is the real data distribution and $q$ is the one estimated from the model. $D_{JS}$ is the Kullback–Leibler (KL) divergence function, shown as Formula (6).

$$D_{KL}(p \| q) = \int_x p(x) \log \frac{p(x)}{q(x)} dx \tag{6}$$

Moreover, use the Wasserstein distance as WGAN's loss function:

$$L(p_r, p_g) = W(p_r, p_g) = \max_{w \in W} \mathbb{E}_{x \sim p_r}[f_w(x)]$$
$$- \mathbb{E}_{z \sim p_r(z)}[f_w(g_\theta(z))] \tag{7}$$

where $f$ is a family of K-Lipschitz continuous function, $w$ is a set of parameters in the network $f_w$. The discriminator model is used to optimize $w$, aims to maximize the loss function.

The WGAN has a better performance than GAN, but still has the drawbacks of vanishing gradient and mode collapse. Hence, Gulrajani et al. [27] proposed WGAN-GP to ameliorate the problems above further. The major modification is imported gradient penalty as a regularizer in the loss function. The updated loss function is shown below:

$$L = \mathbb{E}_{z \sim p_r(z)}[D(G(z))] - \mathbb{E}_{x \sim p_r(x)}[D(s)]$$
$$+ \lambda \mathbb{E}_{\hat{x} \sim p_{\hat{x}}}\left[(\|\nabla_{\hat{x}} D(\hat{x})\|_2 - 1)^2\right] \tag{8}$$

$$\hat{x} = t\hat{x} + (1-t)\boldsymbol{x} \text{ with } 0 \leq t \leq 1 \tag{9}$$

where $\hat{x}$ sampled from $\hat{x}$ and $x$ with $t$ uniformly sampled between 0 and 1. WGAN-GP is superior to WGAN in training speed and stability, and is insensitive to the selection of hyper-parameters. More detailed studies can be found in [27].

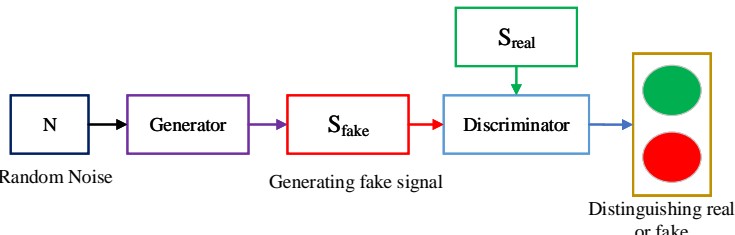

**Figure 1.** GAN sketch map.

## 3. The Framework of the Proposed Method

### 3.1. The Overall Framework

Much of the research up to now [28,29] needs to extract the features manually before training in the area of troubleshooting. That handing is a toilsome work and may lead to losing the time-domain information of the raw signals. In contrast, the DWGANGP has the capability of adaptive feature learning, which can directly recognize the characteristics from the original signals. It consists of two functional networks: the sample generation network (i.e., the WGAN-GP model) and the fault recognizing network (i.e., the CNN model), connecting in series. The flow chart of this model is shown in Figure 2. The core procedures are as follows:

(1) A signal segment with the fixed length will be randomly collected from the raw time-domain signals, aiming to fabricate the fault sample. The step above will be repeated several times to gather the fault samples corresponding number.

(2) The samples of minority class are trained by the sample generation network model to produce the synthetic samples.

(3) The unbalanced real samples are assembled with synthetic samples to augment the samples of the minority class.

(4) The rebalanced fault samples are trained and classified by the fault recognizing network.

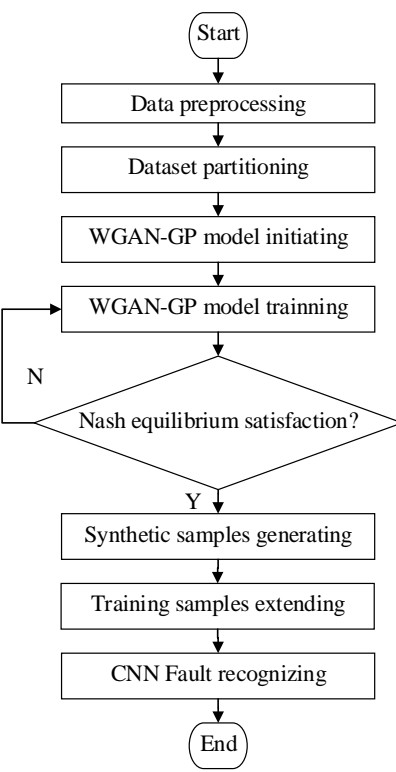

**Figure 2.** The flow chart of the DWGANGP model.

### 3.2. The Networks Architectures

The DWGANGP is coupled by WGAN-GP and CNN, which are the role of the sample generation network and the fault identification network (FIN), respectively. Their detailed structure parameters were shown in Table 1. The discriminator and generator of WGAN-GP are composed of convolution layers and transposed convolution layers. The generator's input is the uniformly distributed random noise, which is reconstituted into synthetic samples after three transposed convolution operations and two up-sampling operations. Besides, its first two convolutional layers use ReLU as the activation function and perform batch normalization, while the last convolutional layer uses Tanh as the activation function, and the output is the synthesized sample. The discriminator takes either a real sample or a composite sample, uses LeakyReLU as the activation function, and the output is a binary number. Both networks mentioned above use RMSProp as the optimizer, with an initial learning rate of 0.00005. The fault identification network consists of a three-layer convolution network. It uses the ADAM as the optimizer, the learning rate is set to be 0.001, and the training iterations is set to be 200.

**Table 1.** Detailed structure parameters of the networks.

| Model | Generator | Discriminator | FIN |
|---|---|---|---|
| Layer1 | Input:$1 \times 100 \times 1$ | Input:$1 \times 400 \times 1$ | Input:$1 \times 400 \times 1$ |
| Layer2 | Upsampling-1:$1 \times 2$ | Conv-1:$1 \times 30 \times 16$ | Conv-1:$1 \times 30 \times 32$ |
| Layer3 | Conv-1:$1 \times 20 \times 128$ | Conv-2:$1 \times 10 \times 32$ | MaxPool-1:$1 \times 2$ |
| Layer4 | Upsampling-2:$1 \times 2$ | Conv-3:$1 \times 10 \times 64$ | Conv-2:$1 \times 10 \times 64$ |
| Layer5 | Conv-2:$1 \times 10 \times 64$ | Conv-4:$1 \times 10 \times 128$ | MaxPool-2:$1 \times 2$ |
| Layer6 | Conv-3:$1 \times 3 \times 1$ | FC-1:Flatten | FC-1:Flatten |
| Layer7 | Output:$1 \times 400$ | Output:1 | FC-2:256 |
| Layer8 | \ | \ | Output:10 |

## 4. Experimental Results Analysis

### 4.1. Dataset Introduction

The data set in this paper is collected from Western Reserve University Bearing Data Center (CWRU) [30]. The test platform is shown in Figure 3. Its test system is composed of the motor, dynamometer and control circuit. The drive end bearings model is SKF6205, which is made with faults using electro-discharge machining. Faults including 0.007 inches, 0.014 inches and 0.021 inches in diameter were presented separately at the inner ring, ball and outer ring. The vibration signals were acquired by the accelerometer for motor loads of 0 to 3 horsepower (HP). In this paper, 1000 real samples were randomly selected as the training set, another 1000 samples as the testing set, and an additional 500 real samples were selected as the validation set. No data are reused among the three sets. According to the unbalanced proportion of data, the corresponding number of training set samples would be deleted, then fill in the generated samples to form the new training set. Case 2 and case 3 inherited the training set in case 1 for training. Samples from the HP = 2 and HP = 3 were used as testing sets and validation sets for case 2 and case 3, respectively. The specific division method is shown in Table 2. Considering that the sampling frequency was 12 kHz, and the motor speed was 1797 rpm in no-load, we determined the sample length to be 400, which approximated the sampling number that bearing rotated by a full turn.

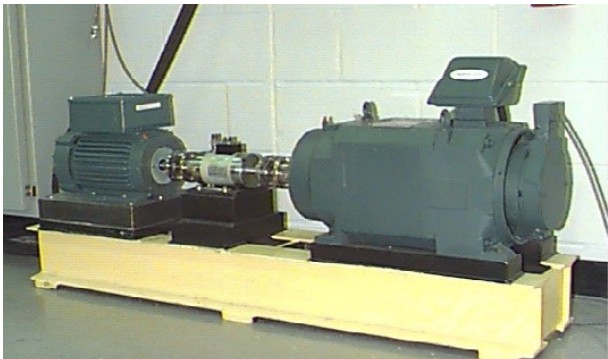

**Figure 3.** The test platform.

**Table 2.** Description of data set.

| Damaged Position | Normal | Ball | | | Inner Raceway | | | Outer Raceway | | |
|---|---|---|---|---|---|---|---|---|---|---|
| Damaged diameter (inches) | \ | 0.007 | 0.014 | 0.021 | 0.007 | 0.014 | 0.021 | 0.007 | 0.014 | 0.021 |
| Class label | 0 | 1 | 2 | 3 | 4 | 5 | 6 | 7 | 8 | 9 |
| Training set (HP = 0) | 1000 | 1000 | 1000 | 1000 | 1000 | 1000 | 1000 | 1000 | 1000 | 1000 |
| Testing set (HP = 0) | 1000 | 1000 | 1000 | 1000 | 1000 | 1000 | 1000 | 1000 | 1000 | 1000 |
| Training set (HP = 2) | 0 | 0 | 0 | 0 | 0 | 0 | 0 | 0 | 0 | 0 |
| Testing set (HP = 2) | 1000 | 1000 | 1000 | 1000 | 1000 | 1000 | 1000 | 1000 | 1000 | 1000 |
| Training set (HP = 3) | 0 | 0 | 0 | 0 | 0 | 0 | 0 | 0 | 0 | 0 |
| Testing set (HP = 3) | 1000 | 1000 | 1000 | 1000 | 1000 | 1000 | 1000 | 1000 | 1000 | 1000 |

### 4.2. Analysis of Training Effects

Figure 4 shows the training performance of the diagnostic networks in 1:50 balance ratio. As Figure 4a shows, the loss curves of training decrease with the increase of iteration numbers. The CNN and BP models show a faster convergence rate, which reaches convergence around the 10th and 50th epoch, respectively. The DBN model shows a relatively slow convergence, demonstrating a stepwise descent and reaching a stable convergence around the 80th epoch. Figure 4b indicates the curves of the validation accuracy between them. The accuracy of all three improved with the increase of training times. The accuracy of the CNN model has a sharp rise in the first 25 epochs and achieves a stale diagnosis accuracy at the 50th epoch. The accuracy of the BP model increases gradually, and achieves

balance at the 85th epoch. Correspondingly, the accuracy of the DBN model shows an interval fluctuation before the 110th epoch, then has a steep increase. Compared with BP and the DBN model, the CNN model shows a better performance in both training speed and diagnostic accuracy in the diagnostic networks.

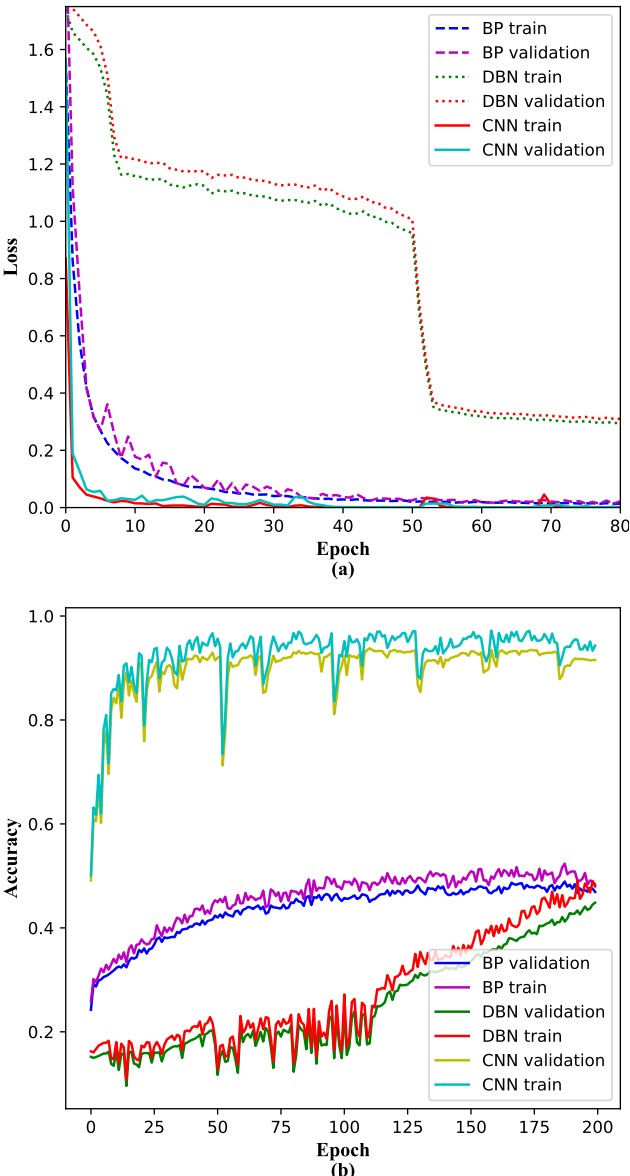

**Figure 4.** The diagnostic networks' training curve in 1:50 balance ratio: (**a**) the loss curve of the training set and validation set (**b**) the accuracy curve of the validation set and training set.

### 4.3. Quality Analysis of Synthetic Samples under Highly Unbalanced Data

The quality of synthetic samples is an essential factor affecting the classification accuracy in the imbalance diagnosis problem. However, compared with the image samples, the one-dimensional signals are often difficult to appraise, because it is hard to evaluate the authenticity, as the image sample does visually. Nonetheless, the t-distributed stochastic neighbor embedding (t-SNE) [31] technique could be used to assess the quality of synthetic samples to a certain extent.

In this experiment, the balance ratio was chosen to be 1:100, and the NR, SMOTE, GAN and WGAN-GP methods were applied respectively to generate the synthetic samples. As the extended samples, the synthetic samples would be reconstructed with the real samples and then be trained by the CNN model. The fully connected output would be

extracted as the input of the t-SNE algorithm. After the calculation of t-SNE, the output of fully connected, which contained high-dimensional features, would be reduced as the two-dimensional data. The visualizing result was displayed in Figure 5. Figure 5a illustrates the distribution of the diagnostic features extracted by the NR-CNN method. Half types of the ten fault characteristics can be well classified, except for class 2 to class 4, which remain overlapping. Moreover, the boundary between class 5 and class 10 is not clearly separated. As shown in Figure 5b, the features that come from the SMOTE-CNN method cannot correctly cluster except class 1. Figure 5c shows t-SNE features extracted by GAN-CNN method. Like the NR-CNN method, class 2, class 3 and class 4 are blended in feature space, and class 8 and class 10 are also not properly divided. Figure 5d demonstrates the distribution of features extracted by the DWGANGP method. It can be visibly presented that most categories are well-separated clusters, except class 2 and class 4. In contrast with the other three oversampling methods, the WGAN-GP model can generate high-quality synthetic signals, which have more adaptation for the CNN diagnostic network.

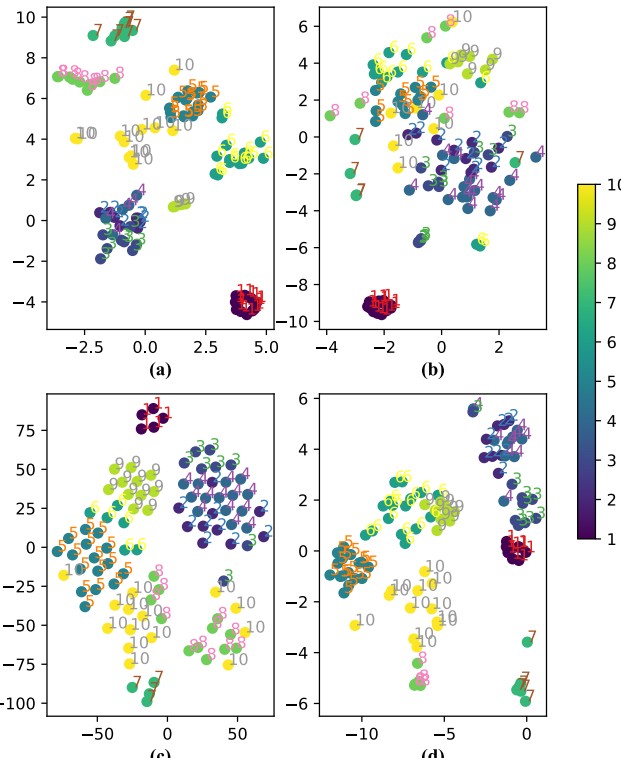

**Figure 5.** The t-SNE visualized feature distribution in 1:100 balance ratio between different oversampling techniques on the CNN model: (**a**) NR; (**b**) SMOTE; (**c**) GAN and (**d**) WGAN-GP.

### 4.4. Analysis of Classification Accuracy

This part proposed a comparative experiment to evaluate different models' performance on the imbalance fault diagnostic problem in three different scenarios. The NR, SMOTE, ADASYN, GAN and WGAN-GP were selected as the sample generators to produce the synthetic samples in different balance ratios. All the synthetic samples were then added into the samples in minority classes, aiming to increase the imbalance samples to balance. Finally, these extended samples were trained separately by the SVM, BP, DBN and CNN diagnostic models. The dataset selection for the three cases was shown in Table 3. The details of partitioning for the dataset were illustrated in Table 4.

#### 4.4.1. Case 1: Imbalance Fault Diagnosis

The curves of diagnostic accuracy were shown in Figure 6. All models' diagnostic accuracy increased with the balance ratio increase and finally approached the balanced

sample result (i.e., the balance ratio is 1:1). Besides, under all balance ratios, the models' diagnostic accuracy is better than that of the no-sampling models. The WGAN-GP method has an enormous improvement in diagnostic accuracy on BP, DBN and CNN models, while it is the NR method in the SVM model. Overall, the diagnostic accuracy of the DWGANGP model is far better than that of all other fault diagnosis models, which has an average leading of 53%, 40% and 46% compared to NR-SVM, WGANGP-BP and WGANGP-DBN.

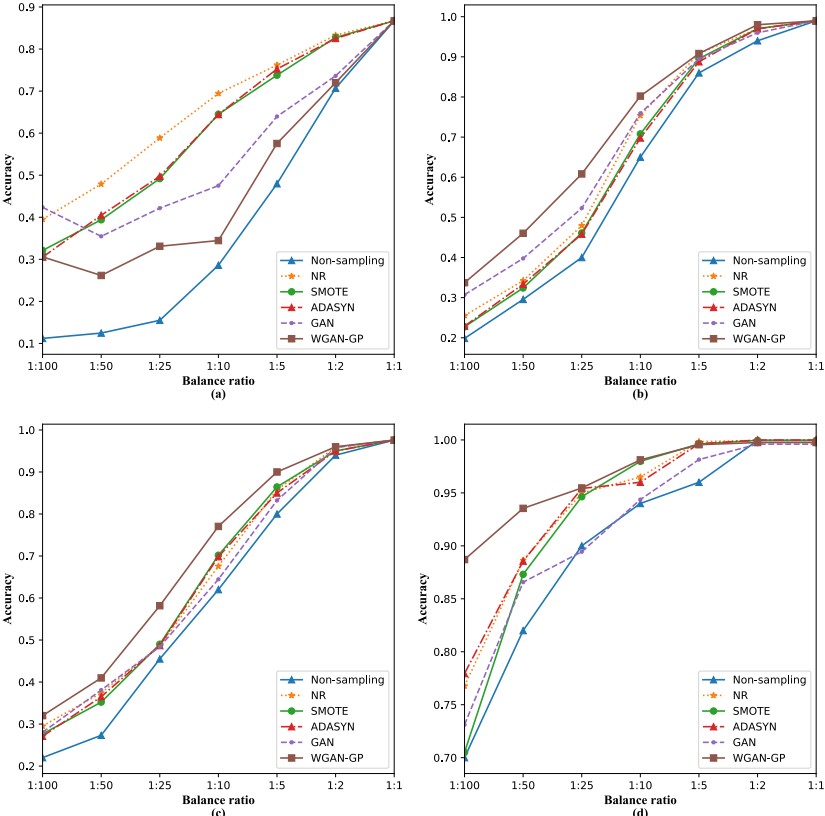

**Figure 6.** Diagnosis accuracy of multi-class imbalance fault dataset on HP = 0: (**a**) SVM model; (**b**) BP model; (**c**) DBN model and (**d**) CNN model.

### 4.4.2. Case 2: Imbalance Fault Diagnosis under Variant Motor Load

In this part, we evaluated the diagnostic accuracy of the models mentioned above under the conditions of motor load–variant, based on the unbalanced samples problem. The partitioning method of the training set was the same as that of case 1. However, the data HP = 2 was used for the testing set instead of HP = 0, aiming to verify these models' generalization. The dataset selection was shown in Table 4, and the precision curve was shown in Figure 7.

Compared with case 1, the diagnostic accuracy of each model decreased to different degrees. Notably, as shown in Figure 7b, the SMOTE and ADASYN methods have no noticeable effect on improving accuracy. The diagnostic accuracy in SVM, BP and DBN models is greatly affected by the condition of variable load, while the influence of the CNN model is relatively small. The WGAN-GP method still has good adaptability in BP, DBN and CNN models, which shows the best performance in the correspond models. Comprehensive analysis shows the DWGANGP has better generalization than other models under the condition of the variable load. Meanwhile, the DWANGP has over 80% in average diagnostic accuracy and has an average leading of 74%, 86% and 114% compared to NR-SVM, WGANGP-BP and WGANGP-DBN.

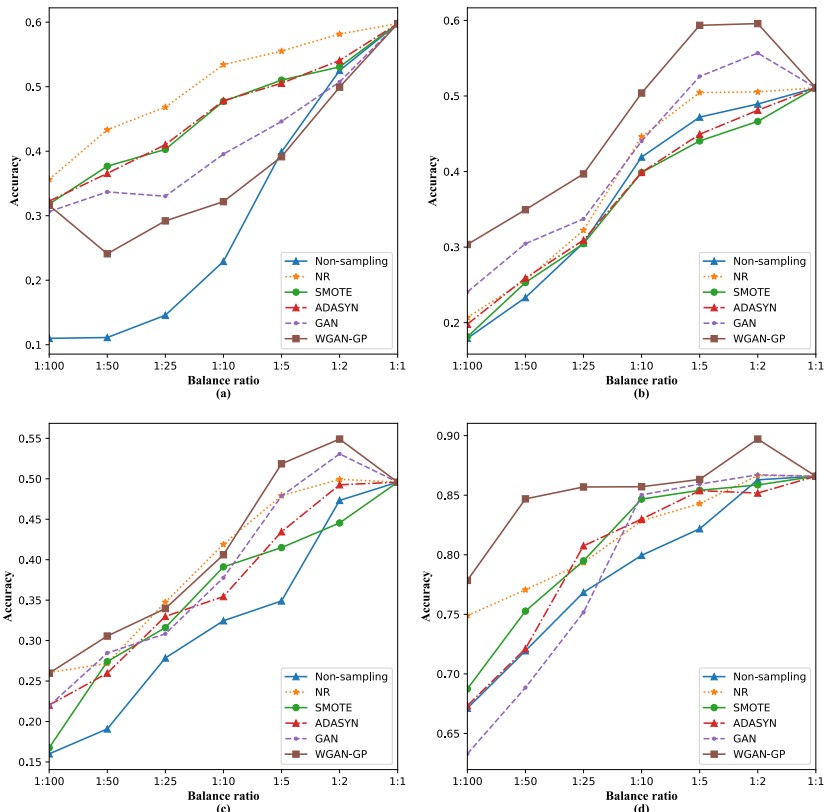

**Figure 7.** Diagnosis accuracy of multi-class imbalance fault dataset under variant Motor load: (**a**) SVM model; (**b**) BP model; (**c**) DBN model and (**d**) CNN model.

**Table 3.** The dataset partitioning for imbalance fault diagnosis under variant motor load and noise.

|  | **Training Set** | **Testing Set** |
|---|---|---|
| Case 1 | HP = 0 | HP = 0 |
| Case 2 | HP = 0 | HP = 2 |
| Case 3 | HP = 0 | HP = 3 with Gaussian noise of SNR = 10 dB |

**Table 4.** The data set partitioning of the of imbalance fault diagnosis for each category.

|  | **Balance Ratio** | **1:100** | **1:50** | **1:25** | **1:10** | **1:5** | **1:2** | **1:1** |
|---|---|---|---|---|---|---|---|---|
|  | synthetic samples number | 990 | 980 | 960 | 900 | 800 | 500 | 0 |
| Training set (HP = 0) | real samples number | 10 | 20 | 40 | 100 | 200 | 500 | 1000 |
|  | balanced samples number | 1000 | 1000 | 1000 | 1000 | 1000 | 1000 | 1000 |
| Testing set (HP = 0) | samples number | 1000 | 1000 | 1000 | 1000 | 1000 | 1000 | 1000 |

### 4.4.3. Case 3: Imbalance Fault Diagnosis under Variant Motor Load and Noise

This section would verify each model's performance under relatively extreme conditions that considered motor variant-load and noise jamming factors simultaneously. The experiment would be designed as follows: In the training part, we would follow the same setup as in case 1, but used HP = 3 instead of HP = 0 as the testing part to create the variant-load condition. Additionally, the white Gaussian noise (WGN) with the specified signal to noise ratio of 10 dB would be mixed with the signals on HP = 3 to simulate the environment of noise. The dataset selection was shown in Table 4 and the detail of accuracy was shown in Figure 8.

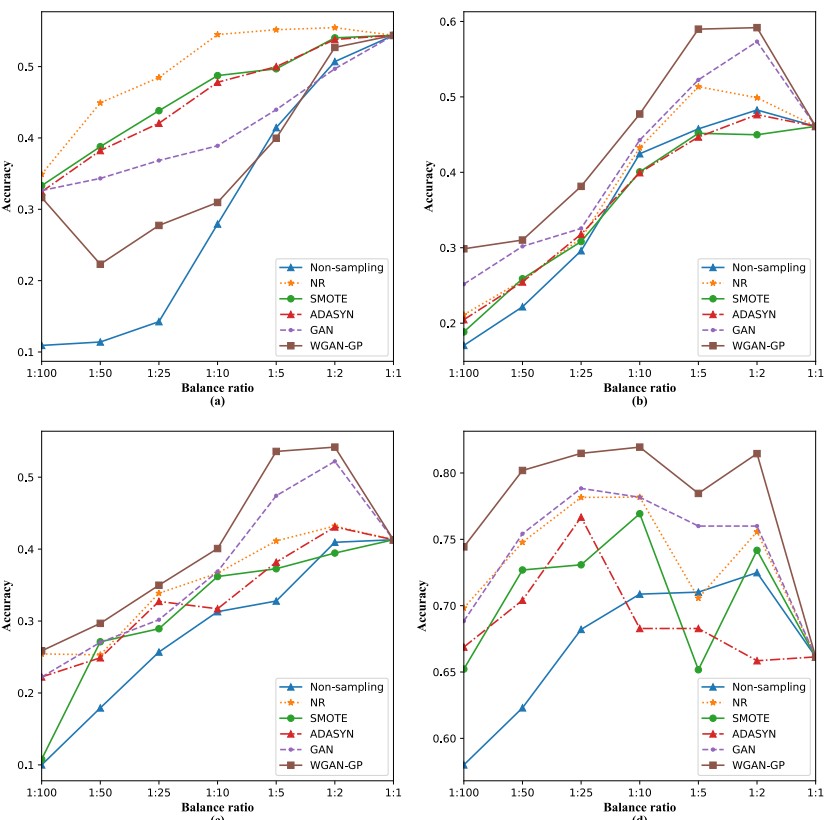

**Figure 8.** Diagnosis accuracy of multi-class imbalance fault dataset on HP = 3 and SNR = 10 dB: (**a**) SVM model; (**b**) BP model; (**c**) DBN model and (**d**) CNN model.

Compared with case 2, each model's diagnostic accuracy is further reduced, and the degree of differentiation is increased. The fault diagnosis network based on CNN is far superior to other models in diagnostic accuracy. The DWGANGP outperforms all other models, which has around 80% in the average diagnostic accuracy and has an average leading of 65%, 81% and 103% compared to NR-SVM, WGANGP-BP and WGANGP-DBN. Remarkably, the DWGANGP model's accuracies in all balance ratios are better than that in the balanced samples.

## 5. Discussion

Through the analysis from case 1 to case 3, it can be summarized that the CNN network has better capacities in feature extraction, anti-noise, and the robustness under variable working conditions, compared with the independent fault recognizing networks. Moreover, all the oversampling methods are effective at solving the unbalanced fault diagnosis problem. In particular, the combination between WGAN-GP and artificial neural networks (ANNs) always get the optimal results, except combined with the SVM model. This result may support the hypothesis that the synthetic samples generated by GAN contain more nonlinear features, which are classified by ANNs easily but not for SVM. WGAN-GP used in conjunction with CNN (i.e., the DWGANGP model) exhibits the finest generalization under load–variant and noise conditions, based on imbalance fault diagnosis. The DWGANGP can produce high-quality synthetic signals in a highly unbalanced ratio to suppress overfitting. Hence, the DWGANGP model can notably improve the diagnostic accuracy in a highly unbalanced ratio.

## 6. Conclusions

This study set out to develop an end-end model for the imbalanced fault diagnosis of rotating machinery. The raw vibration signals can be processed by our model directly. In the WGAN-GP network, the counterfeit samples with similar real signal features can

be generated to extend the minority samples through the discriminator and generator's competitive mechanism. Then, the CNN diagnosis network can accurately classify the reconstructed vibration signals. The comprehensively comparative tasks are designed to consider the variable working conditions and the noise effect. These experiments confirmed that the DWGANGP model is superior in training convergence, diagnostic accuracy under highly unbalanced ratios and generalization. Thus, the model has the potential to be applied in the industrial environment. After a comprehensive analysis of the experimental results, the following conclusions are obtained:

(1) In contrast to SVM, BP and DBN fault diagnostic networks, the CNN model has 80%, 46% and 48% average rises separately on accuracies, revealing a superior features extraction ability for fault diagnosis problems.

(2) DWGANGP models can generate the synthetic sample with similar features of the original samples. The generated samples can be mixed with the original samples to be trained by CNN network, which improves the recognition accuracy under the condition of unbalanced samples.

(3) Under varying speed and noise conditions, the diagnosis accuracies of our method are around 74% and 81% in extremely small and moderate imbalance ratio respectively, showing a strong fault feature extraction capability and robustness.

**Author Contributions:** Conceptualization, J.L. and L.Z.; methodology, J.L.; validation, Q.L., D.L. and M.C.; writing—original draft preparation, review and editing, J.L.; visualization, supervision, project administration, funding acquisition, L.Z. All authors have read and agreed to the published version of the manuscript.

**Funding:** This research was funded by Special Fund for Bagui Scholars of the Guangxi Zhuang Autonomous Region, under Grand 2019A08; The Basic Ability Enhancement Program for Young and Middle-aged Teachers of Guangxi, under Grand 2020KY10018.

**Institutional Review Board Statement:** Not applicable.

**Informed Consent Statement:** Not applicable.

**Data Availability Statement:** Not applicable.

**Conflicts of Interest:** The authors declare no conflict of interest.

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
