# Peer review of "Imbalanced Fault Diagnosis of Rotating Machinery Based on Deep Generative Adversarial Networks with Gradient Penalty"

_processes, doi:10.3390/pr9101751_

Round 1

Reviewer 1 Report

The article describes the solution of the problem of failure classification of rotating machines with the use of machine learning methods. The challenge is to ensure the balance of the training samples that will be used to train the convolutional neural network.
To ensure the balance of the training data, an approach using deep generative networks to generate an artificial training set was proposed.
Then this set was used to train the convolutional neural network. The results were compared with two reference methods, proving the superiority of the CNN-based approach.
Remarks
Figure 4. There is an incorrectly selected scale and the lack of information relevant to the analysis of the learning process of convolutional neural networks. The scale of the loss value is too large, so you cannot see how the CNN learning process has proceeded. The loss function curve is important in the context of detecting the moment of overfitting.
Unfortunately, this is not visible on the graph, there is no course of the loss function on the validation set (there is only the loss on the training dataset). However, the accuracy value on the validation set was given without the training set. There is no information on the breakdown of the dataset. The training and test set (1: 1 division for 1000 samples) was mentioned. Later, information about the validation set appears (e.g. in the aforementioned drawing). In what proportions was the data set divided (training, validation and testing)?
"Also, for the test in case 1, we randomly selected 1000 samples in each category and regrouped them as the training set and test set."- this information is not precise.
Moreover, it is worth answering the question why artificially generated samples were included in the test set? There is no motivation for this approach. The lack of balance in the data is problematic in the training set, not the test set.

Reviewer 2 Report

Dear authors,

I have read with interrest your paper.

In my opnion, this one is easy to read and I have understand the methodoly presented.

Nevetheless, I have several questions about the signal acquisitions.

You wrotte: "Considering the sampling frequency was 12 kHz, and the motor speed was 1797 rpm 155 in no-load, we determined the sample length to be 400, which approximated the sampling number that 156 bearing rotated by a full turn. Also, for the test in case 1, we randomly selected 1000 samples in each 157 category and regrouped them as the training set and test set."

Please, in your opinion, what is the best sampling frequency to use considering the signal to analyze?

In your opinion, what is the best of sample lenght for your case?

In your opinion, can you reduce the number of bearing rotated and consequently the number of category?

I consider your approach as convincing.

Best regards.

Reviewer 3 Report

The subject is relevant to the field of this journal. The authors should make some improvements to the structure of the manuscript to be suitable for publication.

The language should be improved (Moderate English changes required)

The abstract does not provide the reader with numeric values of the results. It needs to be improved, giving more information about efficiency of the proposed method.

Please remove Fig.7 in section “4.4.2. Case 2: Imbalance fault diagnosis under variant Motor load” and Fig.8 in section “4.4.3. Case 3: Imbalance fault diagnosis under variant Motor load and noise”.  

The Conclusions are very poor and are not suitable, should give more useful conclusions. Should include more numeric values for the results. The authors should rewrite more analytically this section.
